# Sunflower Oil as a Renewable Resource for Polyurethane Foams: Effects of Flame-Retardants

**DOI:** 10.3390/polym14235282

**Published:** 2022-12-03

**Authors:** Magdalene A. Asare, Prashant Kote, Sahilkumar Chaudhary, Felipe M. de Souza, Ram K. Gupta

**Affiliations:** 1Department of Chemistry, Pittsburg State University, 1701 S. Broadway Street, Pittsburg, KS 66762, USA; 2National Institute for Materials Advancement, Pittsburg State University, 1204 Research Road, Pittsburg, KS 66762, USA

**Keywords:** bio-based polyol, sunflower oil, epoxidation, expandable graphite, dimethyl methyl phosphonate

## Abstract

Currently, polyurethane (PU) manufacturers seek green alternatives for sustainable production. In this work, sunflower oil is studied as a replacement and converted to a reactive form through epoxidation and oxirane opening to produce rigid PU foams. Confirmatory tests such as Fourier-transform infrared spectroscopy (FT-IR), gel permeation chromatography (GPC), and hydroxyl value among others were performed to characterize the synthesized polyol. Despite the versatility of rigid PU foams, they are highly flammable, which makes eco-friendly flame retardants (FRs) desired. Herein, expandable graphite (EG) and dimethyl methyl phosphonate (DMMP), both non-halogenated FR, were incorporated under different concentrations to prepare rigid PU foams. Their effects on the physio-mechanical and fire-quenching properties of the sunflower oil-based PU foams were elucidated. Thermogravimetric and compression analysis showed that these foams presented appreciable compressive strength along with good thermal stability. The closed-cell contents (CCC) were around 90% for the EG-containing foams and suffered a decrease at higher concentrations of DMMP to 72%. The burning test showed a decrease in the foam’s flammability as the neat foam had a burning time of 80 s whereas after the addition of 13.6 wt.% of EG and DMMP, separately, there was a decrease to 6 and 2 s, respectively. Hence, our research suggested that EG and DMMP could be a more viable alternative to halogen-based FR for PU foams. Additionally, the adoption of sunflower polyol yielded foams with results comparable to commercial ones.

## 1. Introduction

Polyurethanes (PUs) are materials that have a broad reach of applications due to their multifaceted nature. Among their many uses, they can be manipulated into sealants, adhesives, thermoplastics, and flexible/rigid foams depending on their chemical composition [1]. Rigid PU foams (RPUFs) are most suitable in the construction industry for filling and thermal insulation due to their unique properties. They have a low density, high chemical resistance, high compressive strength, and other beneficial thermomechanical properties [2,3]. Rigid PU foams are usually synthesized through an addition reaction of a polyol and diisocyanate in the presence of blowing agents, catalysts, and stabilizers [4,5]. Polyols, which contain hydroxyl groups, influence the mechanical and physical properties of the foams. For example, polyols with a high hydroxyl number show a high apparent density in the foams [5,6]. Yet, the great majority of components used to make polyurethanes are derived from petroleum, and due to its fast depletion along with the need for renewable alternatives, scientists are driven to use bio-based oils as reagents in the synthesis of PU foams [7]. Plant-based alternatives such as soy oil, corn oil, castor oil, limonene, and canola oil are renewable, less costly, easily attainable, nontoxic, and nonvolatile [5,6]. In addition, scientists have found different ways of modifying and converting the unsaturated compounds present in some vegetable oils into hydroxyl groups that are reactive with isocyanate in the formation of a urethane linkage [7,8,9]. Guo and his research team, for example, synthesized a bio-based polyol from soybean oil using epoxidation followed by a ring-opening reaction using methanol [10]. Other successful methods include thiol-ene, transesterification, hydroformylation, and ozonolysis followed by hydrogenation [11,12,13].

Sunflower (*Helianthus annuus* L.) is a majorly produced polyunsaturated oil that has a diverse amount of fatty acids such as linolenic, linoleic, oleic, stearic, palmitic, and arachidic [14]. This diversity permits its numerous applications in foods, lubricants, biofuels, and cosmetics [15]. In addition, it can be used in making medicines and white wine [16]. It is also worth mentioning that Victor and his team used a combination of 15% and 10% of soybean and sunflower oil (SFO), respectively, in PU foam production and their results showed comparable properties to commercial foams [17]. Sunflower oil has a high composition of unsaturated groups containing 14–39% and 48–74% of oleic and linoleic acids, respectively, which makes it a good candidate for the epoxidation/ring-opening reaction to synthesize a bio-based polyol [18]. In the epoxidation reaction of an olefin, an oxygen atom from a peroxide is introduced into the double bond to form an epoxide, which is reactive and can be converted into a hydroxyl group [19]. Successful conversion of an epoxide into a hydroxyl group has been performed through the following: (a) catalytic hydrogenation; (b) a reaction with HCl or HBr, resulting in halogenated polyols; or (c) a ring-opening process through methanol under acidic catalyzed media to obtain a methoxylated polyol, or performing the process with H_2_O to form vicinal –OH moieties [10].

This work employs the epoxidation reaction sequenced by ring-opening using methanol to synthesize sunflower oil-based polyol as this approach is one of the most efficient, greenest, and cheapest ways for the formation of epoxide that is highly reactive and can be converted into polyol [20,21]. Despite the numerous advantages and applications of PUs, they are generally faced with the issue of high flammability, which is usually reduced with the introduction of FRs [22]. Popularly used FRs are halogenated, nitrogenous, or phosphorus-based; however, due to the toxic nature of the halogenated ones, extensive research on the production of non-halogenated FRs is encouraged [23]. In this work, non-halogenated FRs such as EG and DMMP were included in the synthesis of the RPUFs. Previous work has investigated the efficiency of EG and DMMP as FRs in foams and has found them to form protective layers that prevent the further spread of fires within the foams, henceforth improving their FR properties [24,25]. The mechanical, morphological, and structural properties of the sunflower polyol in the foams were investigated. In addition, the effects of the carefully selected FR on the flammability of the bio-based RPUFs were characterized and studied.

## 2. Experimental Details

### 2.1. Materials

Sunflower oil was obtained from a local Walmart (Pittsburg, KS, USA), and DMMP and EG were purchased from Sigma Aldrich (St. Louis, MO, USA) . Jeffol SG-522 (Sucrose polyol with OH# 522) and Rubinate M isocyanate (methylene diphenyl diisocyanate) were gifted by Huntsman (The Woodlands, TX, USA). 1,4-diazabicyclo [2.2.2]octane (DABCO T-12) (T-12), bis(2-dimethylaminoethyl) (NIAX A-1) (A-1), and dibutyltin dilaurate (DBTDL) catalysts were obtained from Air Products (Allentown, PA, USA). Silicon surfactant (Tegostab B-8404) was obtained from Evonik (Parsippany, NJ, USA). Toluene, tetrafluorobic acid (HBF_4_), hydrogen peroxide, acetic acid, Amberlite IR 120H, Lewitt MP64, sodium sulfate, sodium chloride, and methanol were purchased from Fisher Scientific (Allentown, PA, USA). Distilled water, which was employed as a blowing agent was obtained from the local Walmart (Pittsburg, KS, USA).

### 2.2. Synthesis of Sunflower-Based Polyol

#### 2.2.1. Epoxidation of SFO

The procedure adopted in this work: the sunflower oil was epoxidized similarly to previously reported routes [26,27]. An in situ epoxidation reaction of unsaturation: acetic acid: H_2_O_2_ was performed in a 1:0.5:1.5 molar ratio. The SFO, amberlite resin, and toluene were mechanically stirred in a 3-necked flask equipped with a water bath and automatic heating. At a maintained temperature and equilibrium, acetic acid and H_2_O_2_ (30%) were introduced in the reaction in a dropwise manner. After the complete addition of the reactants, the mixture was stirred for several hours at 70 °C. Following, the setup was cooled to room temperature and the amberlite resin was decanted and filtered out. The oil and aqueous layer were separated by gravity along with multiple washing with 10% brine solution. Anhydrous sodium sulfate (Na_2_SO_4_) was used as a drying agent and excess solvents were removed from the epoxidized sunflower oil (ESFO) with a rotary evaporator. The epoxide number for the ESFO was 5.6 EOC%. The iodine value was also performed, which yielded very low values that further confirmed the consumption of double bonds.

#### 2.2.2. Ring Opening of Epoxidized SFO

In the ring-opening reaction, a mole ratio of 7:1 of methanol and epoxide was used. The amount of HBF_4_ used was equal to 50 wt.% of water, plus 0.05 wt.% of methanol and epoxidized oil. The reaction system of methanol and the acid was heated to 70 °C in a three-necked flask attached to a condenser and dropping funnel. After mechanically stirring for several minutes, the previously synthesized ESFO was added in a dropwise manner and the reaction was refluxed for an hour. Lewatit MP 64 ion exchange resin was added to a cooled mixture to neutralize as well as to avoid hydrolysis. The reaction mixture was then filtered to remove the resin followed by rotary evaporation. The synthesized polyol was further characterized to confirm the formation of the hydroxyl group. The schematic of the epoxidation and ring-opening of SFO is presented in Figure 1.

### 2.3. Characterization of Epoxide and Polyol

The synthesized epoxide and polyol were characterized through several standards including ISO (International Organization for Standardization) and ASTM (American Society for Testing and Materials) to confirm their formation. The iodine value was determined by Hanus method and phthalic anhydride pyridine (PAP) was used to calculate the hydroxyl number of the polyol according to the ASTM-D 4274. Following the IUPAC 2.201 standard, an indicator method was used to determine the acid value. The percentage of oxirane oxygen was measured using glacial acetic acid in the presence of tetraethylammonium bromide. FT-IR results were obtained using PerkinElmer Spectrum Two spectrophotometer at room temperature. GPC was executed using a system by Waters (Milford, MA, USA) to investigate the molecular weights of the SFO, its respective epoxide, and polyol. The GPC instrument was composed of four 300 × 7.8 mm phenogel and 5μ columns with varying pore sizes of 104, 103, 102, and 50 Å. Tetrahydrofuran (THF) was used as the eluent solvent at a rate of 1 mL/min at 30 °C. Viscosity was determined through a AR 2000 dynamic stress rheometer (TA Instruments, New Castle, Delaware, USA) at room temperature with shear stress increasing from 1 to 2000 Pa linearly. The rheometer was equipped with a cone plate that had a 2° angle and a 25 mm diameter cone.

### 2.4. Preparation of Rigid Polyurethane Foams

A one-pot method was sufficient in the formation of the RPUFs. Two sets of foams using DMMP and EG were prepared. The combination of sunflower polyol and commercial polyol was a 50/50 *wt./wt.* ratio. The equivalent weight of isocyanate in the formulation was based on the equivalent weight of the sunflower polyol and distilled water using Equation (1) [28]:(1)$$\omega i=E\omega i\left[\;\frac{\omega p}{E\omega p}+\frac{\omega pc}{E\omega pc}+\frac{\omega water}{E\omega water}\;\right]$$

For that, the weights of isocyanate (*ω_i_*), SFO polyol (*ω_p_*), commercial polyol (*ω_pc_*), and water (*ω_water_*) were introduced in Equation (1). Following on that, the equivalent weights of isocyanate (*E_ωi_*), SFO polyol (*E_ωp_*), commercial polyol (*E_ωpc_*), and water (*E_ωwater_*) were also inserted. The hydroxyl equivalent of water is *E_ωwater_* = 9. All RPUFs were made by adding 10 g of synthesized sunflower polyol, 10 g of commercial polyol (SG-522), 0.18 g of A-1, 0.04 g of T-12, 0.4 g of B8404 surfactant, 0.8 g of water. Along with that, increasing amounts of DMMP, or EG FRs were mixed using a mechanical stirrer at high speed to form a uniform mixture. The amounts of DMMP and EG, their weight percentage, and sample codes are presented in Table 1. The effect of DMMP and EG on the flame retardancy of the foams was studied separately by adding increasing amounts to the mixture. After a homogenous mixture was formed, isocyanate was carefully charged, and thoroughly stirred for a few seconds. Lastly, the mixture was allowed to rise at room temperature and the foams were maintained for about a week to cure before performing any tests.

### 2.5. Characterization of Rigid Polyurethane Foams

Standard techniques were used to analyze the thermal physio-mechanical properties of the bio-based polyurethane foams. The foams were cut into desired sizes before proceeding to test. Each sample had three specimens for testing to obtain the average of the results. The densities of the foams were calculated following the standard test procedure for the apparent density of rigid cellular plastics using ASTM D1622. The closed-cell content of the foams was measured with Ultrapycnometer, Ultrafoam 1000 according to ASTM 2856 standard method. Following the ASTM 1621 procedure, dimensions of 50 × 50 × 25 mm^3^ for (L) × (B) × (H), respectively, were used to determine the compressive strength of the various foams using a compression fixture-equipped Q-Test 2-tensile machine (MTS, Norwood, MA, USA). The 10% yield at break was obtained with Blue Hill software following the ISO 844:2016 and a strain rate of 30 mm/min was applied vertically on the samples. Thermal degradation of the foams was measured using thermogravimetric analysis (TGA) on a TA instrument (TGA Q500, New Castle, Delaware, USA). The thermograms were obtained under an ultra-pure N_2_ atmosphere under a ramp increase of 10 °C/min. The foam’s flammability properties were studied through the ASTM D 4986-18 standard test, which involved the direct exposure of the foams with dimensions of 150 × 50 × 12.5 mm^3^ for (h), (l), and (t), respectively, in a horizontal placement against a flame applied perpendicularly for 10 s. During the burning, the total time for the sample to and erase the fire was recorded. The weight of the foams before and after burning was also measured to calculate the weight loss percentage. To investigate the morphology and microstructure of the foams, cube sizes of 0.5 cm^3^ were cut and used for scanning electron microscope (SEM) (Phenom, The Netherlands) imaging. Since the foams were not electrically conductive, a magnetron sputtering with a monitor from Kurt J. Lesker Company (Jefferson Hills, PA, USA) was used to sputter gold on their surface to coat them with a thin conducting layer of gold.

## 3. Results and Discussion

### Characterization of Sunflower-Based Epoxide and Polyol

The synthesized sunflower polyol presented a hydroxyl number of 180 mg KOH/g and a considerably low acid value of 0.37 mg KOH/g. The Fourier-transform infrared spectroscopy results shown in Figure 1a illustrate the transformation and confirmation of the epoxide and polyol from the various reactions. In pure SFO, the 3002 cm^−1^ peak is indicative of the =C–H stretch, which is usually above 3000 cm^−1^ [29] and it is observed to disappear after the epoxidation reaction. In the ESFO, the quaternary carbons of the epoxy ring C–O–C bending at 839 cm^−1^ confirmed the formation of the epoxide from the reaction with H_2_O_2_ and acetic acid. In addition, the appearance of a hydroxyl peak at 3470 cm^−1^ (usually found by other researchers in a range of 3600–3200 for aliphatic and phenolic groups) [30] and the elimination of the epoxide band confirmed the formation of the polyol from the ring-opening reaction with methanol. Moreover, standardized tests such as iodine value, epoxy number, and hydroxyl values were used to confirm the formation of the polyol from the pure SFO. An experimental iodine value of 100 mg I_2_/100 g of oil was recorded in the SFO, and it was almost negligible for the ESFO and SFO polyol as a result of the chemical conversion of the unsaturated groups into an epoxide and finally a hydroxyl group. According to the GPC chromatogram in Figure 1b, the diminished retention time for the SFO polyol (22.2 min), compared to ESFO (22.5 min) and SFO (22.8 min), suggested an increase in the molecular weight of the SFO polyol. Alongside that, a shoulder could be observed at the lowest retention time of around 20.5 min, which could be attributed to the formation of dimers derived from a side reaction between the ESFO and SFO-polyol. Such behavior has been reported in previous studies [25,31,32].

Also, the high viscosity of 0.73 Pa·s of the SFO polyol relative to lower values in the ESFO and SFO was another indication of the increase in molecular weight and conversion of the SFO to a polyol. Alongside that, the increase in viscosity of the SFO polyol in comparison to ESFO and SFO might be attributed to the hydrogen bonding from the hydroxyl groups that were chemically introduced through the epoxidation followed up by the ring-opening reaction.

With the addition of the different flame retardants in increasing concentrations, it was observed that the densities of most foams were within a 30–45 kg/m^3^ range that fell under the industrial application range from about 20–50 kg/m^3^, which made them comparable to commercial rigid foams [33]. As seen in Figure 2a, the density of the foams is observed to increase with the gradual increase in the FRs, with a higher increment in the case of DMMP. However, as shown in other papers for DMMP in RPUFs, the density gradually decreased with the increase in DMMP concentration [3]. Hence, our observation could be a result of a different interaction of the DMMP with the SFO polyol. Based on this observation, it could be suggested that the P=O from DMMP would interact with the H from the urethane linkage (–HN–C(O)–O–), leading to a hydrogen bonding interaction that culminated in the increase in density. The standard deviation was 7.80 kg/m^3^ and 3.83 kg/m^3^ for DMMP- and EG-containing foams, respectively. Based on the closed-cell content (CCC) shown in Figure 2b, the EG-containing foams had a value around 90%, which had some decrease when the highest concentration of 21.6 wt.% EG was added. Based on that, it could be suggested that the FRs maintained a good barrier of airflow after being incorporated into the foams, which could be comparable to conventionally available PU foams. The increasing concentration of EG promoted mild effects on the foam’s CCC by causing a minor variation only when higher concentrations were employed. However, increasing concentrations of DMMP displayed a more notable gradual reduction in the CCC of the foams. The standard deviations for DMMP- and EG-containing foams were 8.03% and 1.90%, respectively.

Scanning electron microscope imaging was performed to understand the microstructure and morphology of the foams including their cell size distribution and microstructure under the influence of DMMP and EG. As shown in Figure 3, the average cell size of the foams did not significantly change with the increasing concentration of EG as the neat foam presented an average of around 237 µm whereas the average for the EG-containing foams was around 273 µm. Along with that, the cell structure maintained a relatively regular spherical shape and without disruptions even after large quantities of EG were added to the polyurethane matrix. This observation suggested that there was an advantageous interaction between the EG and the SFO-based RPUFs. Through that, it is notable that SFO polyurethane was able to properly accommodate the bulky flakes of EG. This aspect could correlate with the mild increase in density, which suggested that the addition of EG into the polyurethane matrix did not jeopardize the foam’s physical properties. On the other hand, as the concentration of DMMP increased there was a gradual increase in the foam’s cell size along with a more irregular cell structure. Based on that, the average cell size for the DMMP-containing foams was around 329 µm. In this sense, it would be expected that an increase in cell size would lead to a decrease in density. However, there was a considerable increase in density after the addition of DMMP. Such a phenomenon could be more likely attributed to the plasticizing effect of DMMP, which may promote a faster curing process that may lead to some degree of foam shrinkage. This effect had a more influential impact on the density rather than the increase in cell size as there was an overall decrease in volume [34,35,36,37].

The overall mechanical properties of polyurethanes can depend on many variables such as shape, the cell size of their microstructure, chemical structure, density, and interaction with filler, among other factors [38,39]. Hence, the compressive strength of the foams containing increasing concentrations of DMMP and EG was analyzed, and their behavior is shown in Figure 4. It was observed that the neat foam presented a yield at a break of around 300 kN/m^2^ (kPa), meaning that before that point the foam would follow Hooke’s Law by presenting an elastic behavior whereas after that point it would display an irreversible compression (plastic deformation). After the addition of a relatively smaller addition of DMMP such as up to 5.57 wt.% (DMMP-3), there was a steady decrease in compressive strength from 300 to 260 kN/m^2^. Such an effect has been previously observed in other reports, and it is likely due to the plasticizing effect that the –P=O displayed on the foam [40,41]. In that sense, there was, perhaps, an interaction between lone electron pairs from the oxygen of –P=O with the H from the urethane linkage (–HN–C(O)–O–), which led to a hydrogen bonding among the matrix and filler instead of matrix and matrix. This could potentially cause a decrease in the interactions between the polymeric segments leading to a deterioration in mechanical properties. Also, this effect was enhanced along with the increase in the concentration of DMMP causing a shift from RPUF’s behavior from relatively brittle to a softer material. The standard deviation for the compressive strength values for the DMMP-containing foams was 45.13 kN/m^2^.

On the other hand, the addition of EG in the foam preserved the initial behavior observed from the neat RPUF. Also, the compressive strength values were maintained nearly constant and close to the value of the neat RPUF of 300 kN/m^2^. It was only after the addition of higher quantities of EG such as for the EG-8 sample (21.6 wt.%) that there was a compromise in the compressive strength at a yield of 250 kN/m^2^. Such behavior could be correlated with the data collected from SEM that presented a relatively constant cell size regardless of the addition of EG. Through that, it is suggested that EG was able to properly disperse within the polyurethane matrix without compromising the physical properties since both density and compressive strength presented minor changes. Yet, despite these variations, the sunflower-based RPUFs presented higher compressive strength when compared to castor oil-based FR foams [42], RPUFs containing reactive FR groups [40], and bio-based RPUFs containing blended EG [25]. The standard deviation for the compressive strength values for the EG-containing foams was 13.75 kN/m^2^.

The thermal properties of the foams with and without FRs were characterized using TGA and derivative thermogravimetric analysis (DTGA) results. This study was performed under N_2_ gas at 10 ℃/min to get more insight into the thermal behavior of the foams. Based on the TGA and DTGA of DMMP shown in Figure 5a,b, there was a two-step thermal degradation observed. The first thermal decomposition process was observed around 120 to 200 °C. However, it is worth noting that the neat foam did not present this degradation step, which suggested that this process was related to the partial vaporization and decomposition of DMMP [3,41]. This process leads to the disruption of –P–O–C– relatively weaker bonds. Through that, radicals can be formed, which aid in the formation of a condensed char that can improve the thermal stability at later stages of thermal degradation [41,43]. Following up, there was a major thermal degradation that occurred at around 300 °C that presented a similar profile for all the foams, suggesting that this process was related to the cleavage of the urethane bond [44,45,46]. Lastly, there was a third degradation step, which was likely the further decomposition of the depolymerized segments of the polyurethane that were converted into gaseous byproducts [44,47,48]. During this degradation, it was notable that foams with a higher concentration of DMMP presented a lower weight loss than neat foam, which suggested an improvement in thermal stability. Because of that, the foams with a higher concentration of DMMP were more likely to have a higher residual char at 700 °C of around 10 to 13 wt.% whereas the neat foam had a char yield of around 5 wt.%.

For RPUFs composed of EG as shown in Figure 5c,d, the thermal degradation occurred in two steps, the first observed at around 300 °C and the second around 480 °C. The first decomposition at 300 °C could be likely associated with the thermal cleavage of the hard segments, and urethane linkage, as previously stated. In addition, there was also the irreversible expansion of EG, which caused the release of H_2_O, SO_2,_ and CO_2_ that occurred around 250 to 300 °C [25,49,50]. The second decomposition was likely related to the further degradation of polyol and isocyanate. Yet, it is worth noting that the foams with a higher concentration of EG presented a considerably lower weight loss, which was likely due to the formation of the carbonaceous char layer that prevented the material from degrading. This led to higher amounts of char residues at 700 °C with the highest values around 20 wt.%. Alongside that, there was a slight shift of the thermal decomposition towards lower temperatures. This process was likely due to the formation of a protective char layer that was able to prevent the degradation of the polyol and isocyanate fragments into gaseous components [25]. Based on these observations, it was noted that there was an overall improvement in thermal stability for the foams after the addition of DMMP and EG, respectively.

To test the effects of DMMP and EG on the flame retardancy of the sunflower-based PU foams, burning tests were performed to measure the burning times and weight losses. The horizontal burning test using the ASTM standard was adopted. The foams were firmly held and exposed to a burning source for 10 s. After that, the flame was removed, and the time required for the sample to extinguish the fire along with its weight loss was recorded. Based on that, the weight loss and burning time plots for all foams are demonstrated in Figure 6. The neat foam presented the highest weight loss of about 50% along with a burning time of around 80 s. After the addition of DMMP and EG, separately, there was a drastic diminishment in both the weight loss and burning time of the foams. In that sense, the optimum amount of DMMP added was 8 g (13.6 wt.%), which led to a weight loss of 1.06% and a burning time of 2 s. Likewise, the optimal amount of EG added was also 8 g (13.6 wt.%) as it displayed a weight loss of 1.65% and a burning time of 6 s. Further addition of FR led to a milder decrease in weight loss and burning time, which was not considerably better than the flame retardancy performance when smaller quantities of FRs were added. Hence, it has been noted that sunflower-based polyurethanes do not require a higher load of FRs to reach their optimal flame retardancy efficiency. Such an aspect is a good indication that these foams can be suitable for scaled production.

The flame retardancy mechanism is an important analysis to elucidate the overall efficacy of FRs against fire. In that sense, there have been several studies in the literature that focused on addressing the phenomena involved in this process [25,40,50,51]. It has been studied that DMMP is a material that releases relatively unstable gaseous radical species such as PO_2_^•^ and PO^•^, which are capable of reacting with highly exothermic fragments that are formed as byproducts during the polyurethane’s combustion [44,52,53]. After that, the phosphorus-based radicals can precipitate in the form of phosphoric acid and go through a dehydration reaction, which can simultaneously decrease the system’s heat as well as form a protective char layer that prevents further degradation of the polyurethane [3,48,54]. The aspect of this char layer can be observed in Figure 7. Hence, it is likely that DMMP has a flame retardancy mechanism that functions in both the gas and solid phase.

The flame retardancy mechanism of EG is relatively different than DMMP, yet it has been demonstrated to be as effective. In that sense, during the irreversible thermal expansion of EG some gases such as SO_2_, CO_2,_ and H_2_O vapor are released. These gases aid in the dilution of reactive gaseous fragments formed during the polyurethane’s combustion. Most important is that EG physically expands, creating a worm-like carbonaceous char layer that is thermally stable and serves as a shield against heat, oxygen, and radicals, which are core components for the propagation of fire [25,49,55,56]. The aspect of these worm-like structures formed after the burning tests can be seen in Figure 7. The photographs of the foams after the horizontal burning tests are shown in Figure 7 for DMMP and EG. As observed, with increasing concentrations of both non-halogenated FRs, the burnt area of the foam drastically reduced as compared to the pure rigid sunflower-based polyurethane foam.

## 4. Conclusions

In this work, our group successfully synthesized polyurethane foams with sunflower polyol while testing the respective effects of non-halogenated FRs, EG and DMMP on their flammability. The effect of adding these FRs provided a mild increase in density for most samples that remained within the 60 to 55 kg/m^3^ commercial range. It was notable that the incorporation of the FRs in the polymeric matrix promoted some variation in the foam’s morphology as there was an overall increase in the cell size and some irregularities in the structure. Yet, no disruption of the cells was observed even after the addition of large amounts of FRs. The effect on compressive strength after the addition of DMMP led to a decrease from 300 to 260 kPa after the addition of up to 5.57 wt.% (DMMP-3). Adding up to 19.1 wt.% (EG-7) maintained the foam’s compressive strength of around 300 kPa, showing minor variance when compared to the neat foam. A considerable improvement in thermal stability was observed after TGA. Finally, the burning test revealed that lower amounts of both DMMP and EG were required to achieve optimal performance against fire. For instance, a load of 13.6 wt.% of DMMP (DMMP-5) and EG (EG-5), separately, presented weight losses and burning times of 1.06% and 2 s and 1.65% and 6 s, respectively. This work demonstrated a viable approach for the processing of sunflower-based rigid polyurethanes that acquired highly effective flame retardancy with DMMP or EG, showing a potential for large-scale production.

## Data Availability

The data presented in this study are available on request from the corresponding author.

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
