# Peer review of "Sunflower Oil as a Renewable Resource for Polyurethane Foams: Effects of Flame-Retardants"

_polymers, 2022, doi:10.3390/polym14235282_

Round 1

Reviewer 1 Report

This manuscript synthesized a polyurethane using sunflower oil as starting reagent. The subject is interesting and the manuscript has been well-written. The manuscript deserves the publication; however, it needs some revisions before final decision.

1. The abstract is lengthy. Please shortened it.

2. Please describe the exact techniques instead of “other wet chemical analyses” in the Abstract.

3. Please provide supporting reference for Eq. (1).

Please provide the standard deviations for the results, for instance, Figure 2 and Figure 4.

4. It is highly recommended to provide the cell size and the cell density results in a figure. Qualitative presentation of the results is not desirable for a research paper.

5. Please provide the strengths in Pa units.

Author Response

This manuscript synthesized a polyurethane using sunflower oil as starting reagent. The subject is interesting and the manuscript has been well-written. The manuscript deserves the publication; however, it needs some revisions before final decision.

  1. The abstract is lengthy. Please shortened it.

Author's Response: Thank you for your comments. The abstract is revised now. The revised abstract is given below:

Currently, polyurethane (PU) manufacturers seek green alternatives for sustainable production. In this work, sunflower oil is studied as a replacement and converted to a reactive form through epoxidation and oxirane opening to produce rigid PU foams. Confirmatory tests like Fourier-transform infrared spectroscopy (FT-IR) gel permeation chromatography (GPC), hydroxyl value among others were performed to characterize the synthesized polyol. Despite the versatility of rigid PU foams, they are highly flammable which makes eco-friendly flame retardants (FRs) desired. Herein, expandable graphite (EG) and dimethyl methyl phosphonate (DMMP), both non-halogenated FR, were incorporated under different concentrations to prepare rigid PU foams. Their effect on the physio-mechanical and fire-quenching properties of the sunflower oil-based PU foams was elucidated. Thermogravimetric and compression analysis showed that these foams presented appreciable compressive strength along with good thermal stability. The closed-cell contents (CCC) were around 90% for the EG-containing foams and suffered a decrease at higher concentrations of DMMP to 72%. The burning test showed a decrease in the foam’s flammability as the neat foam had a burning time of 80 s whereas after the addition of 13.6 wt.% of EG and DMMP, separately, there was a decrease to 6 and 2 s, respectively. Hence, our research suggested that EG and DMMP could be a more viable alternative to halogen-based FR for PU foams. Additionally, the adoption of sunflower polyol yielded foams with results comparable to commercial ones.

  1. Please describe the exact techniques instead of “other wet chemical analyses” in the Abstract.

Author's Response: Thank you for your comments. The abstract is revised now. It is removed from the text to shorten the length but is provided in the main text. The revised abstract is given below:

Abstract: Currently, polyurethane (PU) manufacturers seek green alternatives for sustainable production. In this work, sunflower oil is studied as a replacement and converted to a reactive form through epoxidation and oxirane opening to produce rigid PU foams. Confirmatory tests like Fourier-transform infrared spectroscopy (FT-IR) gel permeation chromatography (GPC), hydroxyl value among others were performed to characterize the synthesized polyol. Despite the versatility of rigid PU foams, they are highly flammable which makes eco-friendly flame retardants (FRs) desired. Herein, expandable graphite (EG) and dimethyl methyl phosphonate (DMMP), both non-halogenated FR, were incorporated under different concentrations to prepare rigid PU foams. Their effect on the physio-mechanical and fire-quenching properties of the sunflower oil-based PU foams was elucidated. Thermogravimetric and compression analysis showed that these foams presented appreciable compressive strength along with good thermal stability. The closed-cell contents (CCC) were around 90% for the EG-containing foams and suffered a decrease at higher concentrations of DMMP to 72%. The burning test showed a decrease in the foam’s flammability as the neat foam had a burning time of 80 s whereas after the addition of 13.6 wt.% of EG and DMMP, separately, there was a decrease to 6 and 2 s, respectively. Hence, our research suggested that EG and DMMP could be a more viable alternative to halogen-based FR for PU foams. Additionally, the adoption of sunflower polyol yielded foams with results comparable to commercial ones.

The info for the wet chemistry test is also provided in the text. Below is the text provided in the revised manuscript.

The synthesized epoxide and polyol were characterized through several standards including ISO (International Organization for Standardization) and ASTM (American Society for Testing and Materials) to confirm their formation. The iodine value was determined by Hanus method and phthalic anhydride pyridine (PAP) was used to calculate the hydroxyl number of the polyol according to the ASTM-D 4274. Following the IUPAC 2.201 standard, an indicator method was used to determine the acid value. The percentage of oxirane oxygen was measured using glacial acetic acid in the presence of tetraethylammonium bromide.

  1. Please provide supporting reference for Eq. (1).

Please provide the standard deviations for the results, for instance, in Figure 2 and Figure 4.

Author's Response: Thank you for your comments. This information is provided now (Ref. 28)

Szycher M, Szycher’s handbook of polyurethanes., 1st ed., CRC Press, New York, New York, 1999

The standard deviation for DMMP and EG-containing foams were 8.03 and 1.90, respectively (For Fig. 2). The standard deviation for the compressive strength values for the DMMP-containing foams was 45.128. The standard deviation for the compressive strength values for the EG-containing foams was 13.748. (For Fig. 4) (attached). 

  1. It is highly recommended to provide the cell size and the cell density results in a figure. Qualitative presentation of the results is not desirable for a research paper.

Author's Response: Thank you for your comments. This information is provided in the revised manuscript. Below is the added text in the revised manuscript.

Figure 3, the average cell size of the foams did not significantly change with the increasing concentration of EG as the neat foam presented an average of around 237 µm whereas the average for the EG-containing foams was around 273 µm…….. On the other hand, as the concentration of DMMP increased there was a gradual increase in the foam’s cell size along with a more irregular cell structure. Based on that, the average cell size for the DMMP-containing foams was around 329 µm.

  1. Please provide the strengths in Pa units.

Author's Response: Thank you for your comments. We have also provided unit of strength into Pa. Below is the revised text.

The overall mechanical properties of polyurethanes can depend on many variables such as shape, the cell size of its microstructure, chemical structure, density, and interaction with filler, among other factors [38,39]. Hence, the compressive strength of the foams containing increasing concentrations of DMMP and EG was analyzed, and their behavior is shown in Figure 4. It was observed that the neat foam presented a yield at a break of around 300 kN/m2 (kPa), meaning that before that point the foam would follow Hooke’s Law by presenting an elastic behavior whereas after that point it would display an irreversible compression (plastic deformation). After the addition of a relatively smaller addition of DMMP such as up to 5.57 wt.% (DMMP-3), there was a steady decrease in compressive strength from 300 to 260 kN/m2 (kPa). Such an effect has been previously observed in other reports and it is likely due to the plasticizing effect that the –P=O displayed on the foam [40,41]. In that sense, there was, perhaps, an interaction between lone electron pairs from the oxygen of –P=O with the H from the urethane linkage (–HN–C(O)–O–) which led to a hydrogen bonding among the matrix and filler instead of matrix and matrix. This could potentially cause a decrease in the interactions between the polymeric segments leading to a deterioration in mechanical properties. Also, this effect was enhanced along with the increase in the concentration of DMMP causing a shift from RPUF’s behavior from relatively brittle to a softer material. The standard deviation for the compressive strength values for the DMMP-containing foams was 45.128.

On the other hand, the addition of EG in the foam preserved the initial behavior observed from the neat RPUF. Also, the compressive strength values were maintained nearly constant and close to the value of the neat RPUF of 300 kN/m2 (kPa). Only after the addition of higher quantities of EG such as for the EG-8 sample (21.6 wt.%) that there was a compromise in the compressive strength at a yield of 250 kN/m2 (kPa). Such behavior could be correlated with the data collected from SEM that presented a relatively constant cell size regardless of the addition of EG. Through that, it is suggested that EG was able to properly disperse within the polyurethane matrix without compromising the physical properties since both density and compressive strength presented minor changes. Yet, despite these variations, the sunflower-based RPUFs presented higher compressive strength when compared to castor oil-based FR foams [42], RPUFs containing reactive FR groups [40], and bio-based RPUFs containing blended EG [25]. The standard deviation for the compressive strength values for the EG-containing foams was 13.748.

Reviewer 2 Report

The work is devoted to the production of practically important construction materials, polyurethane foams, from affordable renewable sources. The topic of the study is undoubtedly relevant, especially in the context of the gradual transition from petroleum products to renewable sources. According to the reviewer's opinion, the work has no serious flaws. The research is well planned, well presented and may be published after minor revisions.

Comments:

1. DABCO is not deciphered when it appears, the abbreviation is used at once (p.2). The amount of catalysts used is not specified.

2. Formulation on page 5, paragraph 2 contains undeciphered abbreviations A-1, T-12.

3. Primary experimental data on the analyses of the corresponding intermediates and products (changes in the concentration of double bonds, epoxy, hydroxyl groups during the process, conversion of reagents) are missing. They can be given either in the paper or in the Supporting Information.

4. The appearance of the second high-molecular-weight mode on the SFO-POLYOL GPC chromatogram is not discussed.

5. The description of Fig. 2 ("all the tested foams had a value greater than 90%") is not consistent with the data in the figure when the amount of FRs is greater than 8 g.

6. The conclusion “Also, the presence of either FRs in the polymeric matrix did not affect the morphology as the cellular structure was maintained regardless of the FRs load” (Conclusion section) do not correspond to the conclusions from figure 3 (“On the other hand, as the concentration of DMMP increased there was a gradual increase in the foam’s cell size along with a more irregular cell structure.”).

Author Response

Reviewer 2

The work is devoted to the production of practically important construction materials, polyurethane foams, from affordable renewable sources. The topic of the study is undoubtedly relevant, especially in the context of the gradual transition from petroleum products to renewable sources. According to the reviewer's opinion, the work has no serious flaws. The research is well planned, well presented and may be published after minor revisions.

Comments:

  1. DABCO is not deciphered when it appears, the abbreviation is used at once (p.2). The amount of catalysts used is not specified.

Author's Response: Thank you for your comments. This information is provided in the revised manuscript. Below is the revised text:

1,4-diazabicyclo[2.2.2]octane (DABCO T-12) (T-12)………. 0.04 g of T-12

  1. Formulation on page 5, paragraph 2 contains undeciphered abbreviations A-1, T-12.

Author's Response: Thank you for your comments. This information is provided in the revised manuscript. Below is the text incorporated:

bis(2-dimethylaminoethyl) (NIAX A-1) (A-1)

1,4-diazabicyclo[2.2.2]octane (DABCO T-12) (T-12)

  1. Primary experimental data on the analyses of the corresponding intermediates and products (changes in the concentration of double bonds, epoxy, hydroxyl groups during the process, conversion of reagents) are missing. They can be given either in the paper or in the Supporting Information.

Author's Response: We have addressed this in the revised manuscript. Below is the text incorporated:

The synthesized sunflower polyol presented a hydroxyl number of 180 mg KOH/g and a considerably low acid value of 0.37 mg KOH/g……..

Moreover, standardized tests such as iodine value, epoxy number, and hydroxyl values were measured to confirm the formation of the polyol from the pure SFO. An experimental iodine value of 100 mg I2/100 g of oil was recorded in the SFO, and it was almost negligible for the ESFO and SFO polyol as a result of the chemical conversion of the unsaturated groups into an epoxide and finally a hydroxyl group………....

Also, the high viscosity of 0.73 Pa.s of the SFO polyol relative to lower values in the ESFO and SFO was another indication of the increase in molecular weight and conversion of the SFO to a polyol…..

The epoxide number for the ESFO was 5.6 EOC%. The iodine value was also performed which yielded very low values that further confirmed the consumption of double bonds.

  1. The appearance of the second high-molecular-weight mode on the SFO-POLYOL GPC chromatogram is not discussed.

Author's Response: Thank you for your comments. We have discussed this in the revised manuscript. Below is the added text:

Alongside that, a shoulder is observed at the lowest retention time of around 20.5 min, which could be attributed to the formation of dimers derived from a side reaction between the ESFO and SFO-polyol. Such behavior has been reported in previous studies [25,31,32].

  1. The description of Fig. 2 ("all the tested foams had a value greater than 90%") is not consistent with the data in the figure when the amount of FRs is greater than 8 g.

Author's Response: Thank you for your comments. We have revised this.

Based on the closed-cell content (CCC) shown in Figure 2b the EG-containing foams had a value around 90%, which had some decrease when the highest concentration of 21.6 wt.% EG was added. Based on that, it could be suggested that the FRs maintained a good barrier of airflow after being incorporated into the foams which could be comparable to conventionally available PU foams. The increasing concentration of EG promoted mild effects on the foam’s CCC by causing a minor variation only when higher concentrations were employed. However, increasing concentrations of DMMP displayed a more notable gradual reduction in the CCC of the foams.

  1. The conclusion “Also, the presence of either FRs in the polymeric matrix did not affect the morphology as the cellular structure was maintained regardless of the FRs load” (Conclusion section) do not correspond to the conclusions from figure 3 (“On the other hand, as the concentration of DMMP increased there was a gradual increase in the foam’s cell size along with a more irregular cell structure.”).

Author's Response: Thank you for your comments. We have addressed this in the revised manuscript. The revised text is given below.

It was notable that the incorporation of the FRs in the polymeric matrix promoted some variation in the foam’s morphology as there was an overall increase in the cell size and some irregularities in the structure. Yet, no disruption of the cells was observed even after the addition of large amounts of FRs.
